# Sensitivity and Adjustment Model of Electrocardiographic Signal Distortion Based on the Electrodes’ Location and Motion Artifacts Reduction for Wearable Monitoring Applications

**DOI:** 10.3390/s21144822

**Published:** 2021-07-15

**Authors:** Fabian Andres Castaño, Alher Mauricio Hernández

**Affiliations:** Bioinstrumentation and Clinical Engineering Research Group-GIBIC, Bioengineering Department, Engineering Faculty, Universidad de Antioquia UdeA, Calle 70 No. 52-21, Medellín 050010, Colombia; fabian.castano@udea.edu.co

**Keywords:** bioelectromagnetism, electrocardiography, instrumentation and measurement, motion artifacts, sensitivity analysis

## Abstract

Wearable vital signs monitoring and specially the electrocardiogram have taken important role due to the information that provide about high-risk diseases, it has been evidenced by the needed to increase the health service coverage in home care as has been encouraged by World Health Organization. Some wearables devices have been developed to monitor the Electrocardiographic in which the location of the measurement electrodes is modified respect to the Einthoven model. However, mislocation of the electrodes on the torso can lead to the modification of acquired signals, diagnostic mistakes and misinterpretation of the information in the signal. This work presents a volume conductor evaluation and an Electrocardiographic signal waveform comparison when the location of electrodes is changed, to find a electrodes’ location that reduces distortion of interest signals. In addition, effects of motion artifacts and electrodes’ location on the signal acquisition are evaluated. A group of volunteers was recorded to obtain Electrocardiographic signals, the result was compared with a computational model of the heart behavior through the Ensemble Average Electrocardiographic, Dynamic Time Warping and Signal-to-Noise Ratio methods to quantitatively determine the signal distortion. It was found that while the Einthoven method is followed, it is possible to acquire the Electrocardiographic signal from the patient’s torso or back without a significant difference, and the electrodes position can be moved 6 cm at most from the suggested location by the Einthoven triangle in Mason–Likar’s method.

## 1. Introduction

One of the main millennium challenges identified by the World Health Organization (WHO), is to increase the health service coverage that currently is around 40%, where Europe and the Americas stand out with the highest level in health services [1]. An important topic to improve is the continuous patients’ monitoring that until the end of the 20th century was carried out only in care centers [2]. Nowadays, in countries of the European Union (EU) and the Organization of American States (OAS) it is possible monitoring the vital signs of patients in places different to care centers, even with patients in their homes while their health status is registered [3,4]. This has been possible through advances in information and communication technologies, which currently allow the remote monitoring of patients as well as medical consultations through digital platforms [5,6].

A medical exam that has been extended to this type of platform is Electrocardiographic (ECG), thanks to its usefulness in potentially life-threatening diseases diagnosis. The information provided by the ECG is so critical that the professionals must take special care in location of the electrodes on the patient’s torso since this can lead to misdiagnosis of heart disease [7]. On the other hand, the movement of the person and the misplacement of the electrodes cause the signal to be contaminated with artifacts [8,9], which has caused misdiagnosis from 17% to 24% of cases [10].

The correct location of the electrodes on the patient’s torso plays a fundamental role in the correct measurement of the physiological signals, and is something that has been treated for decades with the Einthoven triangle as the Mason–Likar’s method proposes [11,12]. This reinforces the idea that a small variation in the location of electrodes on the volume conductor can lead to obtain a distorted signals altering the diagnosis [13,14]. However, wearable devices have been developed to measure ECG in which the location of electrodes on the volume conductor is modified, with the purpose of reduce the size of the devices in such a way that their wear results more comfortable for the people [15,16].

This change in the electrodes position also has the purpose of reducing the noise and artifacts influence, trying to place the electrodes on places with less adipose tissue, which acts as a dielectric that modifies unpredictably the electrical properties of the volume conductor when the subject makes movement, altering the signal waveform [13,15,16]. However, this sacrifices the quality of the ECG signal obtained in the monitoring using these devices due to most of the time these configurations do not follow the Einthoven triangle method producing some distortion in the acquired signal.

For this reason, if it is possible to determine through a computational model the influence that the location of electrodes has on the measurement in the volume conductor, it could be used to determine the distortion level introduced in the ECG signal, allowing that the measured signals can be adjusted to the leads proposed in the Einthoven triangle. In addition, it would be possible to know the position where the signal shows little distortion and obtain a significant reduction of motion artifacts. This model could be used as a pattern for the design of wearable devices that can be used in the measurement of ECG in everyday environments and for home health care.

This paper presents the determination of a sensitivity and adjustment model for the measurement of changes in the ECG signal waveform when it is acquired in different positions of the torso and the back of the subject, making use of a computational model that describes the electrical signal propagation phenomena in biological tissues simulating cardiac depolarization (QRS complex), cardiac repolarization (ST-TU segments) and electric current propagation through the volume conductor. The obtained model is compared with measurements performed in healthy volunteers recorded in conditions of rest and controlled walking movement, with the purpose of determining the distortion introduced by the variation in electrodes’ location over the acquired ECG signal, and performing the fitting of ECG signal acquired from a non-conventional placement respect to the general use placement.

## 2. Materials and Methods

This work presents a sensitivity and adjustment model to compare the ECG signals acquired in different points of volunteers’ torso, the possibility of adjusting an ECG signal acquired in a non-conventional torso location to a standard location defined by Einthoven triangle in Mason–Likar’s method [12], and the distortion introduced in ECG signal waveform by the variation in the location of these points and motion artifacts is also presented.

A database registered for this purpose, composed of real ECG signals from healthy volunteers was used, and a computational model of the electrical behavior of the heart in the volume conductor is presented. This section describes the database, computational model, and comparison measures used to determine the changes in the signals.

### 2.1. Data Recording and ECG Measurement Points

The connection of an electrophysiological signal recording equipment was made to acquire biopotentials in torso and back of 20 volunteers, locating a network of sensors distributed over their torso and back. The distribution of the electrodes had as reference the location recommended by the triangle of Einthoven in Mason–Likar’s method, and according to anthropometric measurements, the position of the other electrodes was shifted 0.125 times the distance between the two clavipectoral triangles on the horizontal axis, and 0.125 times the distance between the upper part of the sternal angle and the lower part of the umbilical region on the vertical axis [17]. These anatomical distances were different for each volunteer, a length of 28.8±1.4  cm on the horizontal axis and 43.2±3.3 cm on the vertical axis in the reference location was estimated for the volunteers. The average displacement distance of the electrodes for each modified position was 3.6±0.1 cm on the horizontal axis and 5.4±0.4 cm on the vertical axis. The distribution of the electrodes over the volume conductor is presented in Figure 1.

The population consisted of 20 healthy male subjects aged 26.3±5.7 and a BMI of 24.4±4.8. Each volunteer was asked to remain at rest for 5 min and then perform a walk movement in a controlled laboratory environment for 5 min while their ECG signals were recorded with a sampling frequency of 250 Hz, in order to obtain information about the dynamics of ECG signal both by the movement of the subject and by the change in the electrodes position on the volume conductor. The ethics committee for human studies of the Universidad de Antioquia approved the register protocol (Approval report 16-59-711).

### 2.2. Computational Model of Heart Electrical Activity

A three-dimensional computational model of anatomical and physiological approximation of the heart electrical activity and its interaction with the volume conductor was modified, to consider the electrical properties of the different structures of the heart and the surrounding tissues (torso, lungs, blood, bone, among others) [18]. From this computational model, the potential difference between different test points on the torso and back surface of the volume conductor was determined as conventionally performed on an ECG [11].

This model comprised a simplified three-dimensional (3D) version of the human body upper trunk, which was divided into a bidomain model. The first domain consisted of the torso, rib cage, lungs and blood chambers inside the heart. The second domain modeled the heart divided into subdomains: sinoatrial node (SAN), atria (ATR), atrioventricular node (AVN), His bundle (HIS), His bundle branches (BNL), Purkinje fibers (PKJ) and ventricles (VTR), since each subdomain had different electrical properties [17]. On the surface of the torso were located the measurement points on the left arm (LA), right arm (RA), left leg (LL) and right leg (RL) which was used as a reference point $\left(V_{GND}=0V\right)$, to obtain the bipolar leads D1 $\left(V_{LA}-V_{RA}\right)$, D2 $\left(V_{LL}-V_{RA}\right)$ and D3 $\left(V_{LL}-V_{LA}\right)$ as indicated by the Einthoven triangle in Mason–Likar’s method [11,12]. In addition, test points were added as shown in Figure 1. Figure 2 shows the 3D model used in the computational model with its respective subdomains.

The volume conductor was analyzed through two physical interaction models; (1) the electric interaction model given by the Laplace equation to model the interaction between the torso, rib cage, lungs and blood chambers (Equation (Equation 1)); and (2) the modified FitzHugh–Nagumo model that evaluates the propagation of electrical impulses on biological tissues to evaluate the electrical interaction between different structures of the heart [18,19,20].
(1)$$\nabla \;\cdot \;\left(-\sigma_{0}\nabla V\right)=0$$
where $\sigma_{0}$ is the electrical conductivity of each subdomain outside the heart (torso, rib cage, lungs and blood chambers).

The modified FitzHugh–Nagumo model is a simplified model of the Hodgkin Huxley equations which describe how electric potentials in biological tissues are initiated and transmitted and has widely used as a reference for the study of electromagnetic phenomena that occur in nervous and heart cells [19,20]. For this model, three dependent variables were defined: $V_{e}$ is the extracellular potential, $V_{i}$ is the intracellular potential, and *u* is the recovery variable that governs the cellular refraction. Equations (Equation 2)–(Equation 4) present the modified FitzHugh–Nagumo model.
(2)$$\frac{\partial V_{e}}{\partial t}-\frac{\partial V_{i}}{\partial t}+\nabla \;\cdot \;\left(-\sigma_{e}\nabla V_{e}\right)=i_{ion}$$
(3)$$\frac{\partial V_{i}}{\partial t}-\frac{\partial V_{e}}{\partial t}+\nabla \;\cdot \;\left(-\sigma_{i}\nabla V_{i}\right)=-i_{ion}$$
(4)$$\frac{\partial u}{\partial t}=ke\left[\frac{\left(V_{m}-B\right)}{A}-du-b\right]$$
where $\sigma_{e}$ and $\sigma_{i}$ are the extracellular and intracellular electrical conductivity, respectively. $\left(V_{m}=V_{i}-V_{e}\right)$ is the potential difference between intracellular and extracellular spaces and $i_{ion}$ is the ionic electric current and is defined by Equation (Equation 5) inside the SAN and Equation (Equation 6) in the other subdomains.
(5)$$i_{ion}=kc_{1}\left(V_{m}-B\right)\left[a-\frac{\left(V_{m}-B\right)}{A}\right]\left[1-\frac{\left(V_{m}-B\right)}{A}\right]+kc_{2}u$$
(6)$$i_{ion}=kc_{1}\left(V_{m}-B\right)\left[a-\frac{\left(V_{m}-B\right)}{A}\right]\left[1-\frac{\left(V_{m}-B\right)}{A}\right]+kc_{2}u\left(V_{m}-B\right)$$
where *a*, *b*, $c_{1}$, $c_{2}$, *d*, *e*, *k*, *A* and *B* are parameters for adjust the electrical properties of the specific cells of each subdomain. It was considered that the torso was electrically isolated, which implies that the electric current remains inside the volume conductor. Additionally, the interaction between the heart and the surrounding tissue was modeled by the condition of $V=V_{e}$ where $V_{e}$ is the extracellular voltage in the walls of the cardiac muscle. The values of the parameters and the initial boundary conditions for each subdomain are presented in Table 1, and the electrical properties of the subdomains are presented in Table 2, these data were extracted from the literature [19,21,22].

### 2.3. Motion Artifacts, Measures of Sensitivity and Comparison

The ECG is a periodic signal that presents a standard waveform defined by a group of peaks and undulations called segments (*P*, *Q*, *R*, *S*, *T*, *U*) [23]. Each of these segments has a unique shape characterized by its amplitude, duration and time of appearance, and should not present significant variation for each person, because they are the activation signals of the cardiac muscle function in the heart, so that when evaluating the different ECG standard waveforms of a person should be similar [21].

The reference function $S_{0}\left(t\right)$ represents a segment of the reference ECG signal taken in one of the leads indicated by the Einthoven model (point 0 in Figure 1), $S_{j}\left(t\right)$ is the signal to be compared that refers to the segments of the ECG signals taken in other points over the torso from the same lead in a time equivalent to a period $\left[0,T\right]$. It is possible to represent the relationship between both signals by the function presented in Equation (Equation 7) [23,24].
(7)$$S_{0}\left(t\right)=a_{j}S_{j}\left(d_{j}t+t_{j}\right)+c_{j};\;a_{j}>0,d_{j}>0$$
where $a_{j}$, $d_{j}$, $t_{j}$, $c_{j}$ are, respectively, the coefficients of amplitude, duration, time of appearance and offset of the signal. If both signals are similar, then $a_{j}≃1$, $d_{j}≃1$, $t_{j}≃0$ and $c_{j}≃0$. If the variation of the baseline is subtracted in the processing step, $c_{j}$ can be omitted in the Equation (Equation 7).

Volunteers were asked to perform walking movement in a controlled laboratory environment while recording the ECG signal in different points on the torso, to obtain information about the dynamics of the signals while it was contaminated by motion artifacts. To evaluate the differences that exist in the artifact in the different measurement points, the coefficients of each segment was determined, this was done from the Ensemble Average (EA) ECG.

#### 2.3.1. Ensemble Average ECG and Model Comparison

The EA ECG method was used to evaluate the distortion of the signal introduced by the change in the position of the electrodes and by the motion artifacts through the measurement of the difference between the coefficients of the segments. This method allows us to find the average pattern of a signal that has a periodically repeated waveform and that appears as its constituent waveform, as is the case of the ECG signal [25].

To find the EA of the ECG signal it was necessary to select a fiducial point on the standard waveform, which was the reference in time to match the different waves, in this case the R-peak was selected. On the other hand, the size in time or in samples that has the standard waveform was determined to perform the partition of the signal, the synchronization through the fiducial points and the averaging of the signals. This time was determined as the elapsed time between two consecutive R-peaks.

Using this method, each signal from the volunteers was partitioned and analyzed, considering that 5 min of signal were analyzed for the three leads in the torso and the back of the subject in seven different positions as shown in Figure 1 and Figure 2. This allowed us to obtain a database of 10,800 ECG signal segments for the analysis, from which the coefficients of each segment that composes the standard waveform were extracted. The coefficients of the signals obtained from the displaced electrodes were compared with the coefficients of the reference signals (point 0 in Figure 1), to determine the distortion introduced by the change in the location of the electrodes.

#### 2.3.2. Dynamic Time Warping and Signal Distortion

The Dynamic Time Warping (DTW) method allows us to calculate the minimum Euclidean distance between each sample of the signal to be compared $S_{j}\left(t\right)$ and each point of the reference signal $S_{0}\left(t\right)$ [26,27]. The method uses two matrices of identical size to perform the calculation, the matrix ${S1}_{\left(m\times n\right)}$ contains *m* copies of the reference signal $S_{0}{\left(t\right)}_{\left(1\times n\right)}$ in the rows, and the matrix ${S2}_{\left(m\times n\right)}$ contains *n* copies of the signal to compare $S_{j}{\left(t\right)}_{\left(m\times 1\right)}$ in the columns. The distance matrix $D_{\left(m\times n\right)}$ is calculated using the Single Dimension Euclidean Distance as shown in Equation (Equation 8).
(8)$$D\left(x,y\right)={\left[S1(x,y)-S2(x,y)\right]}^{2}$$
where 1≤x≤m and 1≤y≤n. Starting in the position (1,1) of D, a cost matrix C is created to store the accumulated distance of the previous column and row, which are calculated with the Equation (Equation 9).
(9)$$C\left(x,y\right)=D\left(x,y\right)+\min\left\{\begin{array}{c} C(x,y-1)\\ C(x-1,y-1)\\ C(x-1,y) \end{array}\right.$$

Finally, the path of minimum distances is found from the cost matrix C, starting at the position (m,n) of the matrix and moving towards the adjacent position of lowest cost until reaching the beginning, these positions are saved and then will be identified in matrices S1 and S2 to create the minimum difference aligned signals S1w and S2w respectively. In this process, it is possible that some samples of the matrix S1 or S2 are repeated to conform the vectors S1w and S2w, which is an index of the difference between both signals and the distortion of the evaluated signal.

Through this method, the distance between the standard waveform of the reference ECG signal and the ECG signal acquired in the positions displaced on the torso of the volunteers at rest was determined. In the same way, the difference between the signals contaminated with motion artifacts was calculated.

### 2.4. Signal to Noise Ratio and Motion Artifacts

To measure the alteration in ECG signal by the presence of motion artifacts, and the difference of these artifacts with respect to the location of the electrodes on the torso, the Signal-to-Noise Ratio (SNR) was used which allowed us to calculate the difference between a test signal and a reference signal quantitatively, both contaminated with motion artifacts. Additionally, it allows us to measure the energy contribution of the noise in the evaluated signal [28,29]. The SNR is calculated from Equation (Equation 10).
(10)$$\mathrm{SNR}[\mathrm{dB}]=10\log\left(\frac{\sum_{i}{\left|x_{r}\left(i\right)\right|}^{2}}{\sum_{i}{\left|x_{p}\left(i\right)-x_{r}\left(i\right)\right|}^{2}}\right)$$
where $x_{r}\left(i\right)$ is the reference ECG signal, and $x_{p}\left(i\right)$ is the test ECG signal, both contaminated with motion artifacts.

### 2.5. Difference Percentage and Similarity Percentage

The difference percentage allows us to know the difference between two experimental measures $E_{1}$ and $E_{2}$, or between an experimental measurement and a prediction derived theoretically [25]. The percentage difference can be calculated from Equation (Equation 11).
(11)$$\%\mathrm{Difference}=\frac{\left|E_{1}-E_{2}\right|}{\frac{1}{2}\left(E_{1}+E_{2}\right)}\times 100$$

The similarity percentage quantitatively shows how similar two characteristics are and can be calculated using Equation (Equation 12).
(12)%Similarity=100%−%Difference

## 3. Results

This section presents the results of the sensitivity and adjustment model to compare between the ECG signals acquired in the reference location proposed by the Einthoven triangle and the ECG signals acquired in displaced positions in the torso and back of the volunteers, in addition to their comparison with the computational model and the influence of motion artifacts on the ECG signal.

### 3.1. Computational Model of Heart Electrical Activity

The implementation of the mathematical model was carried out using finite elements in COMSOL Multiphysics, which allowed obtaining the solution of the Laplace equation derived from the Maxwell equations and the solution of the modified FitzHugh–Nagumo model. The implemented model consisted of a mesh of 29,129 tetrahedral elements and 5415 vertices, with an average quality of 65%. The error obtained in the convergence was $8.8\times 10^{-16}$.

### 3.2. Ensemble Average ECG, Sensitivity and Adjustment Model

The Ensemble Average ECG allowed a complete characterization of the waveform of the segments that defined the ECG signal standard wave, as presented in the Equation (Equation 7). Figure 3 shows the EA ECG of lead D2 taken at the reference points on the torso of a volunteer.

Figure 3a shows the ECG signal acquired from the volunteer, while Figure 3b presents the Ensemble Average of the signal, the solid blue line represents the average waveform of the ECG signal after superimposing the different characteristic waves of the signal represented by the gray lines in the background of the plot, it is found after matching the fiducial point at the QRS peak of all ECG waveforms in the frame. The dashed blue lines represent the standard deviation of the overlap of the signals. From the Ensemble Average ECG, it was possible to determine the amplitude, duration, and time of occurrence of the *P*, QRS and *T* waves of the signal. The volunteer presented a heart rate of 87 bpm at the moment of register and the maximum amplitude of the signal was 1.75 mV.

From the EA ECG, the characterization of each segment that compose the ECG signal was performed, in magnitude ($a_{j}$), duration ($d_{j}$) and time of occurrence ($t_{j}$). Since the standard waveform has a different amplitude and duration for each volunteer, the EA ECG was normalized for the maximum amplitude and maximum duration presented in the waves of each volunteer, in such a way that they could be comparable [25]. Similarly, this characterization can be performed through Ensemble Average ECG for the signals obtained from the computational model in the reference position (point 0).

The average similarity percentage between the signals obtained from the computational model and the signals obtained from the volunteers was 87.88±9.71%, according to the Equations (Equation 11) and (Equation 12). The similarity percentage for each derivation in front and back of the torso is presented in Table 3, where it is observed that there was a similar accuracy for all the derivations.

Table 4 shows the values of the signal coefficients for reference signal’s leads D1, D2 and D3 in the torso and back of the volume conductor analyzed in the computational model.

Table 5 shows the average value and standard deviation of each of the coefficients of the signal’s segments in reference position according to the Equation (Equation 7) for leads D1, D2 and D3 taken in the torso and back of the volunteers.

### 3.3. Dynamic Time Warping and Signal Distortion

One of the main objectives of this article was to find the distortion introduced by the location of the electrodes in points different from those recommended by the Einthoven method, for this the DTW method was used, which allowed us to determine the difference between the signal acquired in the reference position (point 0) and the positions displaced based on the provided information by the EA ECG (Figure 1).

Figure 4 presents the superposition of the signals of the lead D1 measured in different positions, where the distortion introduced by the change in the location of the electrodes is observed. The signals obtained from the computational model are shown in Figure 4a, while the signals acquired from a volunteer are presented in Figure 4b.

Table 6 shows the result of the DTW method applied to the ECG signals of the volunteers in the specified position with respect to the reference position, the result is shown with average and standard deviation.

Table 7 shows the result of applying the DTW method on the ECG signals obtained from the computational model at the specified position with respect to the reference position.

Figure 5 shows the result of applying the DTW method on the ECG signals acquired from the volunteers and the signals obtained from the computational model, in the positions specified in Figure 1 with respect to the reference position (point 0).

In Figure 5, a group of box and whisker plots is presented which represent the distance between the signals acquired in the volunteers, compared between the position indicated on the horizontal axis and the reference position. Additionally, the solid line represents the distance between the signals obtained from the computational model, comparing the indicated position on the horizontal axis with the reference position. In both cases a progressive increase in distance was observed as it moved away from the reference position.

### 3.4. Signal to Noise Ratio and Motion Artifacts

An analysis of the SNR of the signals contaminated with motion artifacts was performed, acquired in the different locations shown in Figure 1 with respect to the signal acquired in the reference position contaminated in the same way with motion artifacts, with the purpose of measure if the influence of the artifacts was reduced by modifying the location of the electrodes.

Table 8 shows the result of SNR measurement on the different positions and for leads D1, D2 and D3 of the signals acquired from the volunteers while performing controlled movement in a laboratory environment.

On the other hand, the comparison of the signals acquired in movement through the DTW method was made, to determine the percentage of variation that the signal with artifact presents when changing the location of the electrodes. Table 9 presents the result of applying the DTW method on the signals acquired in different positions of the torso while the volunteers performed controlled movement in a laboratory environment.

Figure 6 shows the results of the SNR measurement and the DTW method applied to the ECG signals of the volunteers acquired in different locations of the torso while performing controlled movement in a laboratory environment.

## 4. Discussion

This paper focuses on showing the comparison between the signals acquired in different points of torso and back, and how the variation of these points significantly affects the shape of the ECG that is acquired, providing a computational model that determines the distortion introduced in the ECG signal due to the mislocation of the electrodes at the measurement points recommended by the Einthoven method. For this, the ECG signals were recorded to a group of volunteers and a computational model was used, with the purpose of mapping the sensitivity of the lead field in the volume conductor and to observe the distortion on the signals as a first step to perform the adjustment of the ECG signal acquired in a non-conventional position to the reference signal determined by the Einthoven method.

The database size was selected according to the Kruskal–Wallis criterion [30], which determines that to have a statistical significance of 95% in the sample and considering a statistical power of the test of 90% for the evaluation of characteristics in ECG signal, it is necessary to carry out a study with a population of at least 14 subjects. As shown in Section 2, the sample for this study was 20 subjects, which gives this study a statistical significance greater than 95%. Despite this, the sample was restricted to only healthy male subjects in an age range considered as young adults, which may restrict the dynamics of the obtained results. This makes it necessay to carry out a study with a significantly larger sample and in which dynamic criteria such as the gender and age of the volunteers can be considered.

The percentage of similarity presented between the coefficients obtained from the measured signals of the volunteers and from the computational model in the reference position was 87.88±9.71%, as is showed in Table 3, the difference between both can be due to the simplifications made in the computational model, both in the anatomical structure and in the electrical parameters [18]. In spite of this, the model works satisfactorily for the purpose laid out, since it allows us to have an overview of the electrical activity of the heart [24]. In addition, it allows us to analyze the distribution of the current density in the volume conductor and determine the surface potentials as measured in a conventional ECG and determine the difference between the signals acquired in non-conventional locations, delivering approximate results as shown in Figure 4.

The percentage difference between the signals of the same lead in torso and back was 4.95%. Accordingly, it is possible to measure a lead of the ECG signal both in the torso and in the back of the subject by placing the electrodes on the reference points recommended by Figure 1 [24].

As presented in Figure 5, as the distance between the position in which the ECG is measured and the reference position recommended by the Einthoven triangle increases, the difference between each pair of signals increases [11]. There are differences of up to 400 μV in signals that can have an average amplitude of 1.5 mV, which represents an error of 37.50%. This result shows that only position 1 and in some cases position 2 can be reliable for taking an ECG signal without distortion. The above can be observed in Figure 4, where an overlap of the signals is made to make the comparison and show the distortion introduced by the change in location of the electrodes. This model allows the fitting of the signal measured in locations other than that the Einthoven triangle by giving information about the level of distortion introduced in the parameters used in the modeling of each independent segment of the ECG signal.

On the other hand, one of the main reasons for modifying the location of the electrodes in ECG measurement is the possible reduction of noise and artifacts. In spite of this, in the results shown in Figure 6 there is no significant improvement of the signal when changing the position of measurement over torso of the volunteers. In the case of the SNR a tendency to increase in the central positions, indicating a reduction of the artifacts, it is also observed that the dispersion increases, which indicates that the motion artifacts do not behave similarly in patients.

In the evaluation of the DTW method, it is observed that there is a variation close to 20% and that it does not change with the position. In addition, a reduction in the variation of this difference is shown in positions 3 and 4, showing that the artifact in these positions tends to resemble the artifact in the reference position. This can show that moving the electrodes towards the center of the chest would not imply an improvement in the behavior of the artifacts that remains unpredictable [31].

Some wearable devices designed for the measurement of ECG signals modify the measurement points on the torso and present this change as an alternative to improve signal acquisition [15,32]. However, as evidenced in this article, modifying the sensing position of the ECG signal modifies the information it presents in its waveform, making the information acquired by these wearable devices not valid for medical diagnosis without a validated correction. It is possible to use this information to acquire derived vital signs such as heart rate or R-R variability [33], but to monitor the ECG signal it is necessary that the position of the sensors is not modified beyond position 2 as it is presented in Figure 1.

The computational model of the heart electrical activity behavior presents an approximate result to the real behavior of the cardiovascular system despite its simplifications, it is possible to have a better approximation adjusting the model to improve its anatomical approach and considering deformable body models, and considering also, the artifacts produced by the movement.

According to the results obtained in the ECG signal distortion with respect to the change of the measurement position on the volume conductor, it can be assumed that the variation produced in the signal by the effect of the motion artifacts influences the change that occurs in the volume conductor with movement, since the torso is not a rigid body and as movement occurs, the distance between the different points of that volume changes, causing a change in the electrical properties of the volume conductor and a significant variation in the potentials measured at the surface. As a future work, it is possible to implement a computational model that allows us to consider deformable body models, including respiratory frequency and tidal volume effects on ECG distortion and better anatomical approximation. Additionally, it is necessary to consider a larger number of volunteers and increasing variability in the gender, size and age of the volunteers to improve the model response, this research could provide a comprehensive understanding of physiological behavior of the motion artifacts and allowing us to have diagnostic able wearable ECG devices.

## Figures and Tables

**Figure 1 sensors-21-04822-f001:**
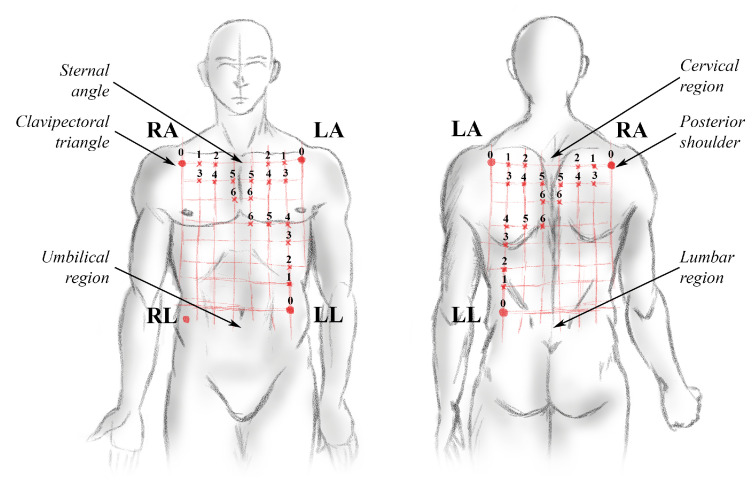
Electrodes distribution for ECG measurement on the torso and back of healthy volunteers with anatomical references. The location of the displaced electrodes is shown with reference to anatomical points recommended by the Einthoven ECG triangle in Mason–Likar’s method.

**Figure 2 sensors-21-04822-f002:**
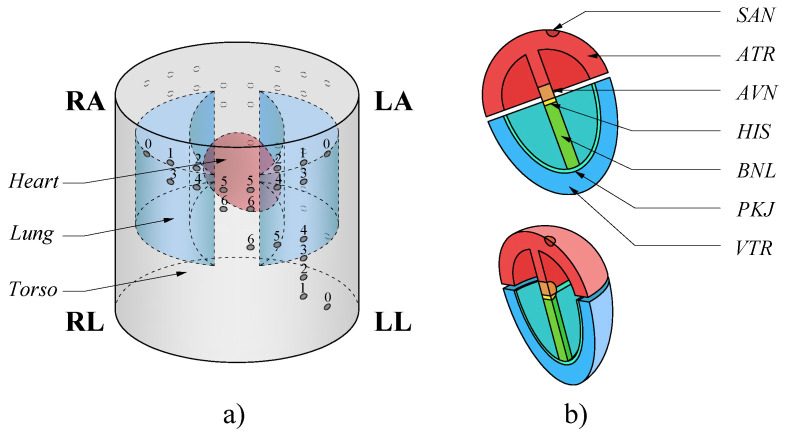
3D model for the volume conductor and heart electrical activity evaluation, the numbers of the electrodes correspond to the positions in which the ECG was evaluated. In (**a**) the subdomains corresponding to torso, lungs and blood chambers are observed. In (**b**) the subdomains that compose the model of the heart are shown.

**Figure 3 sensors-21-04822-f003:**
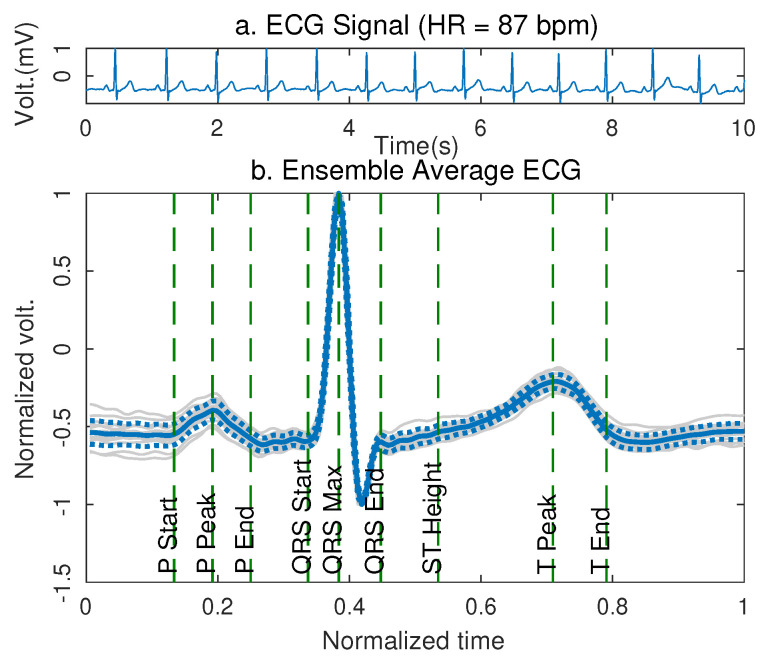
EA ECG of lead D2 of a volunteer acquired at rest in the reference position over the torso (point 0). In (**a**) a 10 s frame of the signal acquired at rest is shown. In (**b**) the EA ECG of the signal is presented with the characterization of the events of the standard waveform.

**Figure 4 sensors-21-04822-f004:**
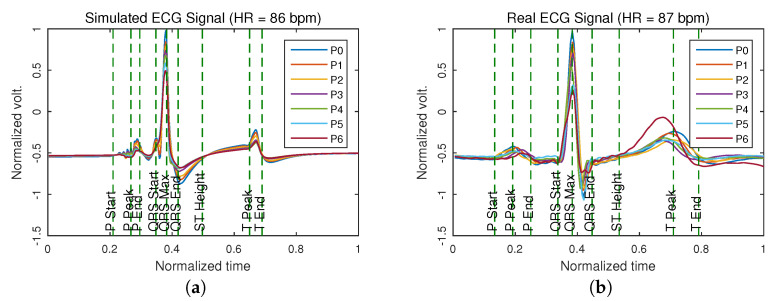
Superposition of the standard waveform of the ECG signal lead D1 measured in different positions of the torso (see Figure 1 for reference). In (**a**) the superposition of the signals obtained from the computational model is presented. In (**b**) the superposition of the signals acquired from a volunteer at rest is shown. Colors indicate the electrodes’ position in the signal acquisition process.

**Figure 5 sensors-21-04822-f005:**
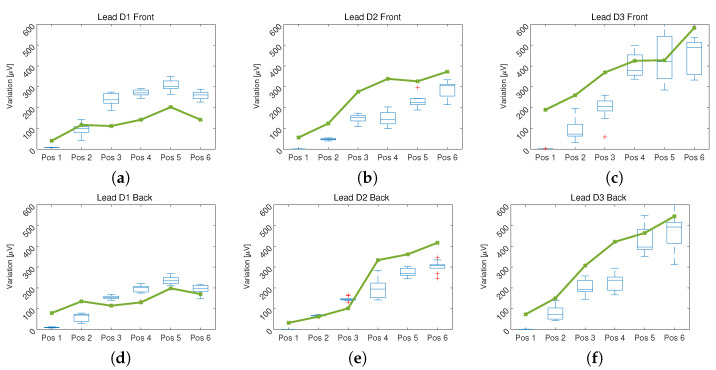
Values of the distance in μV found by the Dynamic TimeWarping method between the ECG signals acquired from the volunteers and obtained from the computational model in the specified position with respect to the reference position: In (**a**–**c**) the result of the measurement of the leads D1, D2 and D3 on the torso respectively is presented. In (**d**–**f**) the result of the measurement of the leads D1, D2 and D3 on the back respectively is presented.

**Figure 6 sensors-21-04822-f006:**
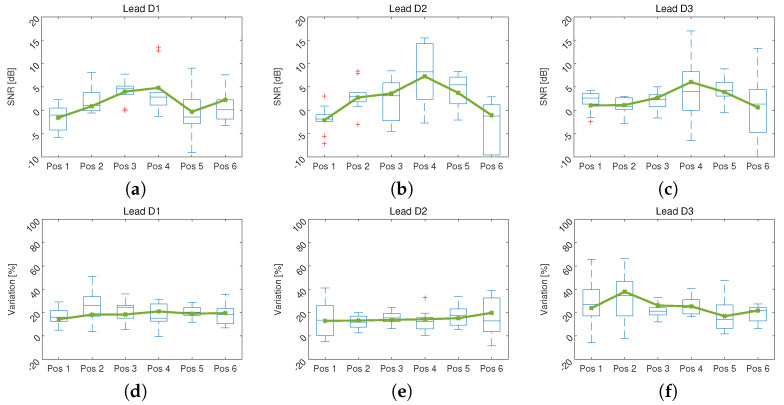
Box and whisker plots of the motion artifacts behavior in different positions of the volunteers’ torso, the continuous line represents the median. In (**a**–**c**) the SNR value of the ECG signal contaminated with motion artifacts in different positions of torso for leads D1, D2 and D3 respectively is shown. In (**d**–**f**) the variation obtained by the DTW method of movement artifacts is presented.

**Table 1 sensors-21-04822-t001:** Values of the adjustment parameters, electrical properties and initial values of the boundary conditions of the model subdomains.

Parameter	SAN *	ATR *	AVN *	HIS *	BNL *	PKJ *	VTR *
*a*	−0.60	0.13	0.13	0.13	0.13	0.13	0.13
*b*	−0.30	0.00	0.00	0.00	0.00	0.00	0.00
$c_{1}\left[AsV^{-1}m^{-3}\right]$	1000.0	2.6	2.6	2.6	2.6	2.6	2.6
$c_{2}\left[AsV^{-1}m^{-3}\right]$	1.0	1.0	1.0	1.0	1.0	1.0	1.0
*d*	0.0	1.0	1.0	1.0	1.0	1.0	1.0
*e*	0.0660	0.0132	0.0132	0.0050	0.0022	0.0047	0.0060
$k\left[s^{-1}\right]$	1000	1000	1000	1000	1000	1000	1000
$\sigma_{e}\left[mSm^{-1}\right]$	0.5	8.0	0.5	10.0	15.0	35.0	8.0
$\sigma_{i}\left[mSm^{-1}\right]$	0.5	8.0	0.5	10.0	15.0	35.0	8.0
$A\left[mV\right]$	33.0	140.0	140.0	140.0	140.0	140.0	140.0
$B\left[mV\right]$	−22.0	−85.0	−85.0	−85.0	−85.0	−85.0	−85.0
$V_{e}\left[mV\right]$ **	0.0	0.0	0.0	0.0	0.0	0.0	0.0
$V_{i}\left[mV\right]$ **	−65.0	−85.0	−85.0	−85.0	−85.0	−85.0	−85.0
*u* **	0.0	0.0	0.0	0.0	0.0	0.0	0.0

* Subdomains of the heart model: Sinoatrial node (SAN); Atria (ATR); Atrioventricular node (AVN); His bundle (HIS); His bundle branches (BNL); Purkinje fibers (PKJ); Ventricles (VTR). ** Initial values of the boundary conditions by subdomain.

**Table 2 sensors-21-04822-t002:** Values of the subdomain electrical properties in the different tissues of the model.

Subdomain	Heart	Blood	Lungs	Muscle	Fat	Bone	Torso
$\sigma_{0}\left[mSm^{-1}\right]$	50 *	700	40 *	200 *	40	6 *	200

* These values are averaged to simplify the model, because they are anisotropic.

**Table 3 sensors-21-04822-t003:** Percentage of similarity between the signals obtained from the computational model and the signals acquired from healthy volunteers (reference position, point 0).

Similarity [%]	$D1_{\mathrm{front}}$	$D1_{\mathrm{back}}$	$D2_{\mathrm{front}}$	$D2_{\mathrm{back}}$	$D3_{\mathrm{front}}$	$D3_{\mathrm{back}}$
Value	89.3±9.2	87.9±12.1	87.0±8.3	88.1±7.8	88.2±8.8	86.9±11.8

**Table 4 sensors-21-04822-t004:** Values of the coefficients of the ECG signal’s segments obtained from the computational model in leads D1, D2 and D3 in torso and back of the volume conductor in reference position (point 0).

Value	$D1_{\mathrm{front}}$	$D1_{\mathrm{back}}$	$D2_{\mathrm{front}}$	$D2_{\mathrm{back}}$	$D3_{\mathrm{front}}$	$D3_{\mathrm{back}}$
$a_{P}$ *	70.3	82.9	97.0	97.2	112.6	109.6
$a_{QRS}$ *	214.6	156.1	873.2	852.4	746.7	784.0
$a_{T}$ *	233.6	165.1	188.7	186.9	178.6	170.6
$d_{P}$ *	86.2	89.1	83.3	92.0	89.1	103.4
$d_{QRS}$ *	71.8	77.6	83.3	83.3	80.5	92.0
$d_{T}$ *	192.5	204.0	212.6	209.8	206.9	198.3
$t_{P}$ *	51.7	51.7	54.6	51.7	40.2	31.6
$t_{QRS}$ *	77.6	63.2	34.5	43.1	43.1	46.0

* Coefficients of the ECG signal’s segments from the computational model: P wave amplitude $\left(a_{P}\left[\mu V/mV\right]\right)$; QRS complex amplitude $\left(a_{QRS}\left[\mu V/mV\right]\right)$; T wave amplitude $\left(a_{T}\left[\mu V/mV\right]\right)$; P wave duration $\left(d_{P}\left[ms/s\right]\right)$; QRS complex duration $\left(d_{QRS}\left[ms/s\right]\right)$; T wave duration $\left(d_{T}\left[ms/s\right]\right)$; time between P wave and QRS complex $\left(t_{P}\left[ms/s\right]\right)$; time between QRS complex and T wave $\left(t_{QRS}\left[ms/s\right]\right)$.

**Table 5 sensors-21-04822-t005:** Average values of the coefficients of the ECG signal’s segments taken in leads D1, D2 and D3 in torso and back of the volunteers in the reference position (point 0).

Value	$D1_{\mathrm{front}}$	$D1_{\mathrm{back}}$	$D2_{\mathrm{front}}$	$D2_{\mathrm{back}}$	$D3_{\mathrm{front}}$	$D3_{\mathrm{back}}$
$a_{P}$ *	92.7±4.5	92.3±3.8	68.5±3.2	67.0±2.6	71.2±3.1	67.4±2.8
$a_{QRS}$ *	544.0±8.1	558.5±10.1	738.9±14.2	750.6±14.6	735.0±14.8	743.7±14.7
$a_{T}$ *	207.3±12.0	170.2±9.6	93.1±4.4	93.7±5.6	108.3±6.9	99.0±5.7
$d_{P}$ *	95.3±1.6	107.7±1.5	105.5±1.7	105.1±1.8	103.9±1.5	101.8±2.0
$d_{QRS}$ *	105.9±2.6	105.0±1.7	101.2±1.8	103.1±1.9	101.2±1.6	108.9±1.8
$d_{T}$ *	214.6±2.5	227.5±5.1	248.2±3.9	243.1±3.7	238.7±3.9	242.1±7.4
$t_{P}$ *	93.1±1.2	99.3±1.9	98.3±1.9	97.5±2.6	96.9±1.9	103.7±2.8
$t_{QRS}$ *	89.6±2.6	97.0±3.0	79.1±2.4	80.3±1.9	73.5±1.6	88.4±2.3

* Coefficients of the ECG signal’s segments: P wave amplitude $\left(a_{P}\left[\mu V/mV\right]\right)$; QRS complex amplitude $\left(a_{QRS}\left[\mu V/mV\right]\right)$; T wave amplitude $\left(a_{T}\left[\mu V/mV\right]\right)$; P wave duration $\left(d_{P}\left[ms/s\right]\right)$; QRS complex duration $\left(d_{QRS}\left[ms/s\right]\right)$; T wave duration $\left(d_{T}\left[ms/s\right]\right)$; time between P wave and QRS complex $\left(t_{P}\left[ms/s\right]\right)$; time between QRS complex and T wave $\left(t_{QRS}\left[ms/s\right]\right)$.

**Table 6 sensors-21-04822-t006:** Average values and standard deviation of the distance in μV found by the DTW method between the ECG signals acquired from the volunteers in the specified position with respect to the reference position for leads D1, D2 and D3 in torso and back.

Dist. [μV]	Position 1	Position 2	Position 3	Position 4	Position 5	Position 6
$D1_{\mathrm{front}}$	8.1±0.9	88.7±27.1	223.7±38.5	271.7±20.2	301.7±24.4	263.5±21.1
$D1_{\mathrm{back}}$	9.8±1.8	59.0±26.6	152.5±14.9	184.8±17.9	239.3±20.4	196.1±23.6
$D2_{\mathrm{front}}$	1.0±0.6	48.6±5.2	145.8±20.8	162.6±34.1	233.1±38.7	292.0±36.1
$D2_{\mathrm{back}}$	0.9±0.3	67.8±4.8	140.0±10.1	186.1±39.2	270.7±23.7	313.8±39.6
$D3_{\mathrm{front}}$	1.8±0.3	94.1±50.3	223.2±71.9	389.9±65.6	446.2±90.0	488.9±92.6
$D3_{\mathrm{back}}$	0.8±0.5	64.3±54.1	189.2±74.4	243.6±63.2	381.6±95.4	453.0±99.7

**Table 7 sensors-21-04822-t007:** Values of the distance in μV found by the DTW method between the ECG signals obtained from the computational model in the specified position with respect to the reference position for leads D1, D2 and D3 in torso and back of the volume conductor.

Dist. [μV]	Position 1	Position 2	Position 3	Position 4	Position 5	Position 6
$D1_{\mathrm{front}}$	40.30	116.41	111.05	141.46	202.18	141.45
$D1_{\mathrm{back}}$	79.07	135.30	114.52	130.26	197.82	170.27
$D2_{\mathrm{front}}$	55.97	123.44	275.74	338.18	326.10	372.51
$D2_{\mathrm{back}}$	32.15	61.68	101.80	333.50	361.46	417.04
$D3_{\mathrm{front}}$	189.31	259.32	368.67	424.79	427.49	583.20
$D3_{\mathrm{back}}$	72.94	150.58	307.16	421.65	463.91	544.57

**Table 8 sensors-21-04822-t008:** SNR value of signals contaminated with motion artifacts acquired in different positions on torso of the volunteers while controlled movement was done.

SNR [dB]	Position 1	Position 2	Position 3	Position 4	Position 5	Position 6
D1	−1.62±3.12	0.84±4.71	3.96±3.95	4.80±5.18	−0.33±5.82	2.19±4.31
D2	−2.14±3.22	2.72±2.54	3.58±4.83	7.25±7.58	3.72±3.62	−1.08±3.89
D3	1.03±3.71	1.08±1.87	2.68±2.03	6.03±5.04	3.89±3.17	0.59±6.80

**Table 9 sensors-21-04822-t009:** Variation of the ECG signal contaminated with motion artifacts in the different measurement positions on torso of the volunteers, using the DTW method.

Var. [%]	Position 1	Position 2	Position 3	Position 4	Position 5	Position 6
D1	14.3±4.1	18.2±10.8	18.4±10.7	21.1±7.3	19.1±8.6	19.6±10.5
D2	12.9±11.9	13.2±7.5	13.8±6.5	14.3±7.1	15.3±10.8	19.8±13.2
D3	23.8±14.3	37.9±25.3	26.1±7.1	25.3±11.1	16.9±12.1	21.7±6.6

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
