# Peer review of "Sensitivity and Adjustment Model of Electrocardiographic Signal Distortion Based on the Electrodes’ Location and Motion Artifacts Reduction for Wearable Monitoring Applications"

_sensors, 2021, doi:10.3390/s21144822_

Round 1

Reviewer 1 Report

Summary recommendation:

             The manuscript titled “Sensitivity and Adjustment Model of Electrocardiographic Signal Distortion based on the Electrodes‘ Location and Motion Artifacts Reduction for Wearable Monitoring Applications” proposed a sensitivity and coordination model for measuring changes in ECG signal waveforms acquired at different locations on the torso and subject's back using a computational model describing electrical signal propagation.

             The manuscript is well organized and the authors' intentions are clear. But before publication, I think that the authors need to answer a few questions and need minor revisions. Detailed suggestions are listed below.

Comment #1:

It seems that the meaning of the Ensemble Average ECG graph in Figure 3(b) is not clear. A detailed explanation of Figure 3(b) mentioned in Section 3.2 should be added.

Comment #2:

The signal analysis of lead D1 measured at different positions in Fig. 4(a),(b) was confirmed. However, the individual waveforms in the color-coded graph are visually ambiguous. It is recommended to show the meaning of each of the classified waveforms through the index box.

Comment #3:

For the contents of Fig. 4 and Fig. 5 in the results section, it is more effective to present information about the reliability and accuracy or correlation between the simulated data and the actually measured data for a clear comparison.

Reviewer 2 Report

The manuscript presents a volume conductor evaluation and an Electrocardiographic signal waveform comparison when the location of electrodes is changed, to find a electrodes’ location that reduces distortion of interest signals.
It was found that it is possible to acquire the Electrocardiographic signal from the patient’s torso or back without a significant difference, and the electrodes position can be moved 6 cm at most from the suggested location by the Einthoven triangle in Mason-Likar’s method.

The paper is well strucutured and written, the methodology is detailed and the results support the conclusions, however I have the following concerns: 
- Keywords are not in alphabetical order.
- No estimation of error is provided for distance used in the electrodes.
- I would like authors to discuss if a larger sample size would be expected of presenting different results.
- The further suggested work should be more detailed.
- There are 5 autocitations, there are all required?
- Only 6 out of 32 references are of the last 5 years.
